# Assessment of Fish Quality Based on the Content of Heavy Metals

**DOI:** 10.3390/ijerph19042307

**Published:** 2022-02-17

**Authors:** Joanna Łuczyńska, Renata Pietrzak-Fiećko, Aleksandra Purkiewicz, Marek Jan Łuczyński

**Affiliations:** 1Department of Commodity and Food Analysis, Faculty of Food Science, University of Warmia and Mazury in Olsztyn, 10-726 Olsztyn, Poland; renap@uwm.edu.pl (R.P.-F.); aleksandra.purkiewicz@uwm.edu.pl (A.P.); 2Department of Ichthyology, Hydrobiology and Ecology of Waters, The Stanisław Sakowicz Inland Fisheries Institute in Olsztyn, 10-719 Olsztyn, Poland; mj.luczynski@infish.com.pl

**Keywords:** heavy metals, marine fish, freshwater fish, Target Hazard Quotient, the combined risk, Hazardous Index

## Abstract

The aim of this study was to estimate the fish quality in terms of the Cu, Fe, Mn and Zn contents. The research material was the muscle tissue of the fish crucian carp (*Carassius carassius* Linnaeus, 1758), flounder (*Platichthys flesus* Linnaeus, 1758), Gilthead seabream (*Sparus aurata* Linnaeus, 1758), mackerel (*Scomber scombrus* Linnaeus, 1758), Blue grenadier (*Macruronus novaezelandiae* Hector, 1871), rainbow trout (*Oncorhynchus mykiss* Walbaum, 1792), tench (*Tinca tinca* Linnaeus, 1758), tilapia (*Oreochromis niloticus* Linnaeus, 1758), Walleye pollock (*Gadus chalcogrammus* Pallas, 1814) and perch (*Perca fluviatilis* Linnaeus, 1758.). Heavy metals were determined with the atomic absorption spectrometry method (AAS). Significantly high concentrations of zinc (19.52 mg/kg wet weight), copper (0.77 mg/kg) and iron (6.95 mg/kg) were found in the muscles of crucian carp (*p* < 0.05) compared to the other fish studied, whereas Walleye pollock had a higher content of manganese (0.266 mg/kg) (*p* < 0.05). All studied fish species do not pose a threat to humans from these four metals. This was indicated by quality indexes (THQ and HI) whose values were below one. The values of these metals also did not exceed the maximum allowable concentrations established by the FAO (1983), but monitoring both the aquatic environment and the fish living there is necessary, for example, for the time-changing abiotic and biotic factors that can cause an increase in metals in the organs of fish.

## 1. Introduction

Fish are an important component of the healthy human diet because of their high nutritional quality. The recommended fish consumption is at least two times a week (quantities of approx. 300 g). The mean daily fish consumption in 2019 in Poland accounted for 12.7 kg per person (244 g/week), and in 2020, it was 13.33 kg per person (256 g/week), while the average person, living in Europe, consumed 24.4 kg of fish or seafood per year [1,2]. Sea fish account for the greatest share in consumption (app. 78%) in Poland. The shares of freshwater fish and seafood are much lower (app. 18% and 4%, respectively) [3].

Fish is one of the main sources of easily digestible protein rich in essential amino acids, fats, macro- and trace elements, and fat-soluble vitamins. Fish is a food rich in valuable long chain polyunsaturated omega-3 fatty acids [4,5].

The appropriate quantities of n-3 PUFA, such as eicosapentaenoic (EPA, C20:5 n-3), docosahexaenoic (DHA, C22:6 n-3) and ocosapentaenoic acid (DPA, C22:5 n-3), prevent or reduce the risk of cancer, cardiovascular diseases and neurological disorders [6,7,8,9]. Scientific studies have confirmed that PUFA play an important role in the growth of the fetus and the development of cognitive functions in children [10].

However, fish also have the ability to accumulate trace elements, heavy metals, pesticide residues and persistent organic pollutants in their tissues, including polychlorinated biphenyls (PCBs) [11,12,13]. The organic pollutants, next to heavy metals, can be harmful to the aquatic ecosystem and humans by the consumption of fish. For example, Georgieva et al. [14] studied the short-term effects of different concentrations of two pesticides (chlorpyrifos (CPF) and cypermethrin (CYP)) in common carp (*Cyprinus carpio* Linnaeus, 1758) under laboratory conditions. The authors, based on the results obtained in the experiment, concluded that, to avoid any risk, these pesticides should be used with caution, especially near water reservoirs. Organic substances like pesticides are harmful to fish, which can be said to reduce the health properties of fish meat [15]. According to Betts et al. [16], the practice of using pesticides is a threat not only for inland fisheries, but it is a global problem, because as a consequence, it can be harmful for human health.

Fish accumulate substantial amounts of metals in their tissues, especially in the muscles. Many factors influence the metal contents in fish tissues, such as environmental quality, season, fish species, stage and age of maturity [12]. According to Nyeste et al. [17], the specific diet of age groups of a given species and the bioindicator capacities of different age groups are also important. Fish samples are considered as one of the most indicative factors, in freshwater systems, for the estimation of the trace metal pollution potential.

Microelements such as zinc (Zn), iron (Fe), manganese (Mn) and copper (Cu), also called trace elements, are present in the human body in amounts less than 0.01%, and the daily requirement is usually less than 100 mg/day [18]. These elements, present in physiological concentrations, are necessary for the proper functioning of living organisms. They are components of many enzymes and take part in many life processes. They are involved in the synthesis of hormones and other substances, helping to regulate the growth, development and functioning of the reproductive and immune systems [19,20,21,22]. Trace elements can cause harmful health effects when they accumulate in organisms in concentrations above those required for metabolic functions. A high consumption of copper, zinc and lead has been linked to Alzheimer’s disease. Zinc and iron have been linked to Parkinson’s disease, and cadmium can cause, among others, kidney dysfunction, osteomalacia and reproductive deficiencies [23,24].

The levels of the trace elements in fish are important, because fish is an important source of food for the general human population [25].

The aims of this study were: determining the copper, iron manganese and zinc contents in ten selected species of fish obtained from the Polish market; conducting a health risk assessment using the Target Hazard Quotient (THQ), the combined risk (TTHQ) and Hazardous Index (HI) and the coverage of the demand (%) for Cu, Fe, Mn and Zn after consuming an average portion of fish (150 g).

## 2. Materials and Methods

### 2.1. Sampling and Sample Preparation

The fish samples: crucian carp (*Carassius carassius* Linnaeus, 1758) (*n* = 6), flounder (*Platichthys flesus* Linnaeus, 1758) (*n* = 6), Gilthead seabream (*Sparus aurata* Linnaeus, 1758) (*n* = 6), mackerel (*Scomber scombrus* Linnaeus, 1758) (*n* = 6), Blue grenadier (*Macruronus novaezelandiae* Hector, 1871) (*n* = 9), rainbow trout (*Oncorhynchus mykiss* Walbaum, 1792) (*n* = 12), tench (*Tinca tinca* Linnaeus, 1758) (*n* = 6), tilapia (*Oreochromis niloticus* Linnaeus, 1758) (*n* = 5), Walleye pollock (*Gadus chalcogrammus* Pallas, 1814) (*n* = 6) and perch (*Perca fluviatilis* Linnaeus, 1758) (*n* = 10) were bought from the local market.

Among the studied fish (Table 1), there were also those from Baltic catches, i.e., Polish Catch Area (flounder); aquaculture Polish inland waterways (rainbow trout); harvested by fishermen from our lakes (perch, tench and crucian carp); fish from ocean catches (Walleye pollock and Blue grenadier—Pacific Ocean and mackerel—Atlantic Ocean) or sea (Gilthead seabream—Mediterranean Sea) and fish imported from China (tilapia).

Whole and gutted fish were transported to the laboratory, and then, the muscle tissue was collected from the dorsal part. The samples were closed in plastic bags and were kept at −30 °C until the analysis. After defrosting the individual parts, they were subjected to grinding and, then, thorough mixing prior to analysis. Frozen fish fillets were thawed and ground just before weighing. A plastic knife and fork were used to prepare the fish. These activities were performed on plastic, disposable plates. Samples were prepared in two parallel replications. After grinding, the samples were weighed and placed in quartz dishes. Reagent tests were performed in parallel.

### 2.2. Element Analysis (Copper, Iron, Manganese and Zinc)

In order to mineralize the samples of whole and gutted fish muscles and frozen fillets, they were dried at 105 °C and then carbonized on plates and incinerated in electric furnaces (Nabertherm GmbH, Lilienthal, Germany) at 480 °C to obtain the color of ash. The obtained ash was diluted in 5 cm^3^ of 1-M nitric acid (Suprapur, Merck, Darmstadt, Germany), then quantitatively transferred with deionized water (Merck-Millipore Elix Advantage 3, Merck-Millipore, Burlington, MA, USA) into 25 cm^3^ flasks. The contents of copper, iron, manganese and zinc were measured with flame atomic absorption spectrometry (iCE 3000 Series AAS, Thermo Scientific, Waltham, MA, USA) [26]. Measurements were conducted at the wavelength 324.8 nm for copper, 248.3 nm for iron, 279.5 nm for manganese and 213.9 nm for zinc, respectively. Four blanks and four standards were analyzed with each batch of samples. The calibration curves were developed using four solution standards (1000 μg/L) with 0.1-M HNO₃ supplied by J.T. Baker^®^ (J.T. Baker Chemicals Company, Deventer, The Netherlands). The calibration curves were linear within the range of heavy metal contents (regression coefficients R^2^ ≥ 0.999). The detection limits (LOD) were 0.05 mg/kg for Cu, 0.5 mg/kg for Fe, 0.05 mg/kg for Mn and 0.1 mg/kg for Zn. The sensitivity was as follows: 0.05 mg/dm^3^, 0.02 mg/dm^3^, 0.02 mg/dm^3^ and 0.05 mg/dm^3^, respectively. The lyophilized certified material (BCR CRM 422 (muscles of cod *Gadus morhua* (L.)) was also analyzed with a known elemental content. The recovery rates were: 103.0% Cu, 96% Fe, 103% Mn and 105.0% Zn, respectively.

### 2.3. Noncarcinogenic Target Hazard Quotient (THQ)

When the THQ < 1, there are health benefits from fish consumption [27,28], whereas a THQ > 1 suggests a high probability of an adverse risk to human health.
THQ = (EFr × ED × FiR × C/RfD × BW X TA) × 10^−3^(1)
where:Efr—the exposure frequency (365 days/year); ED—the exposure duration (70 years); FiR—the fish ingestion rate (g/person/day); C—the average concentration of mercury in foodstuffs (μg/g wet weight); RfD—the Oral reference dose (mg/kg/day) of Zn, Cu, Fe and Mn (RfD = 3.00 × 10^−1^, 4.00 × 10^−2^, 7.00 × 10^−1^ and 1.4 × 10^−1^) (US EPA 2017); BW—the average body weight of local residents (60 kg) [29];TA—the average exposure time (365 days/year × ED).


### 2.4. The Combined Risk of Many Metals

The TTHQ of heavy metals for individual foodstuff calculated as follows [30]:

TTHQ individual foodstuff = THQ (toxicant 1) + THQ (toxicant 2) + ……+ THQ (toxicant n)THQ(Zn) + THQ(Cu) + THQ(Fe) + THQ(Mn)(2)

The HI was calculated using the following pattern:HI = TTHQ (foodstuff 1) + TTHQ (foodstuff 2) + TTHQ (foodstuff 3) + TTHQ (foodstuff 4) + TTHQ (foodstuff 5) + TTHQ (foodstuff 6) + TTHQ (foodstuff 7) + TTHQ (foodstuff 8) + TTHQ (foodstuff 9) + TTHQ (foodstuff 10)(3)

When the HI > 1, there may be a concern for potential health risks [30].

### 2.5. Statistical Analysis

The statistical analyses included calculations of the mean value and the standard deviations, performed with Microsoft Excel software (Microsoft 365, Microsoft Corporation, Redmond, WA, USA). Normal data distribution was analyzed by Shapiro–Wilk’s test, whereas the variance homogeneity was analyzed by Levene’s test. Due to the nonfulfillment of the requirements for the normal distribution of data and variance homogeneity, all the data was then analyzed by nonparametric Kruskal–Wallis, and Dunn’s post-hoc tests were used to determine the statistical differences. The significance of difference of the mean values between the samples was determined with Statistica 13.3 (StatSoft, Inc., 2300 East 14th Street, Tulsa, OK, 74104, USA). Statistical significance was set at *p* < 0.05. The results were expressed on a wet weight basis.

## 3. Results

Table 1 shows the results concerning the contents of heavy metals (Cu, Fe, Mn and Zn) in the muscles of 10 fish species. Among the studied species of fish, significant high concentrations of zinc (19.52 mg/kg), copper (0.77 mg/kg) and iron (6.95 mg/kg) were observed in the muscle tissue of crucian carp (*p* < 0.05), whereas Walleye pollock had a higher content of manganese (0.266 mg/kg) (*p* < 0.05). 

Moreover, the following homogeneous groups were determined (*p* > 0.05).

Copper (Cu): (Blue grenadier, mackerel, rainbow trout, flounder, Gilthead seabream, perch and tench) and (tilapia and Walleye pollock).

Iron (Fe): (mackerel, rainbow trout, tench and perch); (Blue grenadier, Walleye pollock, tilapia and flounder) and (Gilthead seabream).

Manganese (Mn): (crucian carp) and (tench, perch, rainbow trout, tilapia, flounder, Gilthead seabream, Blue grenadier and mackerel).

Zinc (Zn): (rainbow trout, flounder, Gilthead seabream, tench, Walleye pollock, perch, tilapia, mackerel and Blue grenadier).

Table 2 presents the results of covering the daily requirements for selected minerals (Cu, Fe, Mn and Zn) after consuming an average portion of fish (150 g) for six age groups: A—boys 13–18 years old, B—girls 13–18 years old, C—men 19–50 years, D—women 19–50 years old, E—men 51–75 years old and F—women 51–75 years old.

According to the standards developed by the Food and Nutrition Institute for individual macronutrients, differences in the demands for nutrients between the six groups discussed were noted. Boys and men have higher demands for manganese and zinc, while girls and women have a higher demand for iron. As for the demand for copper, it was the same for the six discussed age groups.

In terms of covering the daily requirement after consuming an average portion of fish (150 g), crucian carp is distinguished from other species of fish. One portion of this species of fish covers the highest degree of demand for Cu (12.9%), Mn (0.8–1%), Fe (5.8–10.4%) and Zn (26.6–36.6%). These values are several times higher than in the case of covering the demand after consuming other species of fish. In the case of mackerel and Blue grenadier, the second-highest coverage was recorded after consuming a serving of 150 g per Cu (5.9%). A 150-g portion of tench covers 6.7–9.2% of the demand for zinc. The lowest degree of the demand for Cu is covered by a portion of tilapia (2.7%), of mackerel for Mn (0.1–0.2%), of Gilthead seabream for Fe (0.6–1.2%) and of mackerel and Blue grenadier for Zn (4.1–5.7% and 4.1–5.6%). 

From a nutritional point of view, the fish obtained in this study are safe for the consumer, as the fish quality indicators were below one. The THQ, TTHQ, TDHQ, and HI are presented in Table 3. The crucian carp had the highest THQ index for Zn, Fe and Cu, whereas the highest THQ for Mn was found in the muscles of Walleye pollock. The TTHQ was also the highest in crucian carp muscles.

## 4. Discussion

It is known that fish are a rich source of many nutrients, including macronutrients necessary for the proper functioning of living organisms. Some of them are classified as heavy metals, and their excess poses a risk of human disease. The current study focused on checking the quality of commercial fish in terms of the zinc, copper, iron and manganese contents. Prashanth et al. [31] stated that these metals are essential, because they play an important role in biological processes. Among vertebrates, fish are unique, as they have two routes of metal acquisition: from water via the gills and from the diet via the gut (direct and trophic uptake routes). The direct uptake route is more important, because the gills are the main target organ for metal toxicity in fish [32]. Ali and Khan [33] and Garai et al. [34] reported on the role of fish in biological systems. Their article also mentioned that heavy metals may enter the fish body directly from the water and sediments, through the gills/skin and from its food/prey through the digestive tract. According to Jovanović et al. [35], organisms living in the aquatic environment absorb metals directly from the environment and contaminated water and food, then accumulate them in their tissues and introduce them into the food chain, which is a problem for humans.

The current research concerned the analysis of fish popular on the market and popular for consumers in terms of safety for the consumer. The supplies of fish to the Polish market come from sources such as aquaculture inland waterways, sea fishing and mainly imported. Vietnamese pangasius has been present on the Polish market since 2006. The second species that entered the Polish market was tilapia. The most frequently chosen species are: pollock, cod, carp, trout, pangasius and salmon. Salmon, begins a list of the five most favorite fish, followed by cod, mackerel, carp and tuna [36], The consumer choices in the market for fish, seafood and its products against the background of the situation in the fishing industry [37]. However, it is known that research fish are popular and often eaten around the world. The ecological role of fish should not be forgotten. Ali and Khan [33] reported that the pollution of freshwater ecosystems and fish is a serious issue of environmental, ecological and economic importance. We must not forget about other aquatic ecosystems and the fish that live there. The ecological role of fish was also highlighted by Villéger et al. [38]. The authors wrote that, due to the increasing anthropogenic pressure in freshwater and marine ecosystems, our research should focus on the functions that are related to their main ecological roles in aquatic ecosystems. For most aquatic ecosystems, fish are prey to many other animal species (including other fish), and therefore, their diversity and biomass affect the aquatic predator demographics and, more generally, the structure of aquatic food chains (from pure herbivores to top predators, including various levels of omnivory and detritivory) and adjacent terrestrial ecosystems through the predation of terrestrial animals. It is therefore important to characterize fish defense strategies, as these functional traits affect species adaptation, the structure of fish communities and the functioning of aquatic ecosystems. Winemiller [39] reported that there is a special variation in ecological relationships between different species, as well as different populations within a given species. Aquatic and estuary ecosystems are very sensitive to pollution and landscape changes caused by human activity, as a result of which, fish abundance and diversity have declined in many regions of the world. Taking into account the fact that practically each of the studied species (Table 1) occurs in a different ecosystem, we cannot determine the exact ecological relationships between the described species. The differences between these species may result from many factors, both abiotic and biotic. We have marine and freshwater fish here. There are farmed fish in the studied group (rainbow trout and tilapia) fed with composed feed that are not exposed to such a degree to contact with heavy metals. There are also interspecies differences caused, among other things, by the diet and their place in the food chain. We have nonpredatory fish, which eat mainly bottom food (crucian carp and tench), as well as predatory fish (perch, mackerel, Walleye pollock and Blue grenadier) (Table 1). All these factors influenced the observed concentrations of heavy metals in the muscle tissue of the tested fish. The fact that abiotic and biotic factors influence the contents of heavy metals has been mentioned in earlier publications by other authors [33,40,41,42,43,44,45,46]. 

According to the authors’ knowledge, there are no acceptable standards in Poland regarding the contents of elements in fish. Hence, it was decided to compare the recorded levels of the elements with the regulations contained in Table 4.

On this basis, it was concluded that the levels of Cu, Zn, Fe and Mn in the studied fish species were lower than the maximum levels (Table 1). Bobrowska-Korczak et al. [52] found a that health risk assessment due to contamination is necessary; therefore, monitoring plays a vital role in food safety, which can help introduce national legislation and global standards aimed at reducing or even eliminating exposure to contaminants. 

According to Rožič et al. [53], the Zn and Cu values in cultured and wild seabream were below the permissible levels (50 and 20 mg/kg, respectively) [49]. The values given by the authors were higher than the values determined for this species of fish covered by the research in this paper (Table 1). The muscles of tench caught in the summer from the Damsa Dam Lake included the following levels of Zn, Fe, Cu and Mn: 36.0323, 47.304, 0.5146 and 0.8655 mg/kg wet weight, respectively. These values were much higher than those presented in this study but were safe to be consumed (Table 1) [54]. Seabream from the Sinop Coast of the Black Sea (Turkey) also had higher contents of Zn (10.72–22.34 mg/kg) and Cu (3.48–5.21 mg/kg), but these values were within the limits set by the Commission Regulation and Turkish Food Codex [55]. The muscles of flounder from the Baltic Sea (Poland) contained Zn and Cu values as follows: 14–27 mg/kg dry weight and 0.3–1.1 dry weight [56]. According to Perugini et al. [57], the contents of Cu and Zn in the muscles of Atlantic mackerel were 1.32 and 38.52 mg/kg wet weight, whereas, in the muscle tissue of perch from Anzali Wetland (Iran), they were 10.02 and 27.76 mg/kg wet weight [58]. These values were also higher than those found in the current study for this species of fish (Table 1). The current study also showed significant differences in the contents of copper and manganese in the muscles of seabream and tilapia. Such differences were not noted for zinc (Table 1), while Elnabris et al. [59] did not find statistical differences in the contents of these elements in these fish species. The same authors stated that the average daily intake of metals decreased as follows: Zn > Mn ≈ Cu and was 43.2–239.4, 4.4–9.7 and 2.9–10.6, µg/day/person, respectively. The average daily fish intake in Turkey is 20 g per person, while the EDI values (µg/day/70 kg body weight) calculated by Türkmen et al. [60] were 296.0 for Zn, 22.0 for Cu, 1028 for Fe and 21.2 for Mn and were far below the recommended limits. Hence, consuming these fish did not pose a threat to human health. Abubakar et al. [61] noted a much higher iron content in the muscles of *Scomber scombrus* from Nigeria (11.453 mg/kg and 21.873 mg/kg) and found that these values were above the recommended safety limits provided by the FAO/WHO. Rubio et al. [62] found that a daily average consumption of 45.8 g of Gilthead seabream from Tenerife fish farms (Spain) provided 0.29 mg of Fe (1.6–3.6% of the RDA), 0.062 mg of Cu (6.89–8.86% of the RDA) and 0.83 mg of Zn (7.58–10.43% of the RDA). According to the RDA (Recommended Dietary Allowances) standard, the daily requirement for Cu is 0.9 mg/day in all analyzed age groups. The demand for Mn ranges from 1.6 mg/day (girls 13–18 years old) to 2.3 mg/day (men 19–50 years and 51–75 years old). The daily requirement for Fe is dependent on sex and age, and it ranges from 10 mg/day (men 19–50 years old and men and women 51–75 years old) to 18 mg/day (women 19–50 years old). The daily Zn requirement ranges from 8 mg/day (women 19–50 and women 51–75) to 11 mg/day (boys 13–18 years old, men 19–50 years old and men 51–75 years old) [63]. Individual types of fish meet the highest demand for the discussed minerals among girls aged 13–18 for Mn (0.2–2.5%), men aged 19–50 and men and women aged 51–75 (1.2–10.4%) for Fe (1.2–10.4%) and women aged 19–50 (5.6–36.6%) and 51–75 years (5.6–36.6) for Zn, due to the lowest demand of this age group for these macronutrients.

Many of the heavy metals are micronutrients and trace elements necessary for the proper functioning of the body, e.g., copper and zinc, and only after exceeding a certain level in the body can they cause a toxic effect and interfere with the absorption of other elements.

Budjono and Hasbi [64] studied six important commercial fish species, including *Oreochromis niloticus*, and found that the muscles of all fish meet the limits of Zn for human consumption. According to Bobrowska-Korczak et al. [52], crucian carp contained more Cu than perch and flounder (crucian carp ≈ perch > flounder), although there were no statistically significant differences between crucian carp and perch. This was not in accordance with the authors’ studies (Table 1).

For rainbow trout and other fish species, it was confirmed by Zapata et al. [65] that zinc is present in the muscle tissue, followed by iron and copper. Kalyoncu et al. [66] studied the metal contents in the muscles of fish and observed the following sequences: Zn > Mn > Fe > Cu (tench) and Zn > Fe > Mn > Cu (crucian carp) (Turkey), which were inconsistent with current study (Table 1) and other authors [67]. Fidan et al. [68] examined the contents of heavy metals in crucian carp muscles and noted the same order as in the current study. Yousif et al. [69], based on the research of other investigators, observed that the elements were arranged in the following order: Fe > Zn > Cu > Mn. The findings included fish and other aquatic organisms from polluted areas (rivers and the Karachi Coast, Pakistan). In the muscles of Gilthead seabream purchased from the Bulgarian market, the predominant metal was Zn, followed by Fe, Cu and Mn [70]. This was in accordance with the results in the current study (Table 1). A previous study [45] also confirmed the same order of the elements in the muscles of tench and perch, while Rakocevic et al. [44] found that this sequence was as follows in perch muscles: Zn > Fe > Mn > Cu.

The total target hazard index in wild and farmed seabream from coastal Algeria was less than one [71]. Similarly, the consumption of both farmed species (seabass and Gilthead seabream) from four Mediterranean fish farms should be considered safe due to the contents of the metals for human health [72]. The THQ values for the muscles of cultured Gilthead seabream from the Mediterranean (Corsican Coast) were also lower than 1 (0.103 for Zn, 0.002 for Fe, 0.004 for Cu and 0.001 for Mn in terms of a 70-kg adult consuming 427 g of fish/week [73]. The THQ (0.0002–0.0186) and individual foodstuff TTHQ (0.132–0.653) for the muscles of roach, bream (*Abramis brama* (L.)), pike (*Esox lucius* (L.)) and Eurasian perch were below 1, where mercury was also taken into account in calculating the TTHQ [30]. Similar results were achieved when examining the muscle tissues of European perch and roach from Lake Pluszne [74]. These results were confirmed by the current study (Table 3).

## 5. Conclusions

The crucian carp is considered a tasty and economically valuable fish. The research showed that, among other fish species bought in stores, crucian carp contained the highest levels of heavy metals, which, despite their name, are needed for the functioning of living organisms. The quality indicators used in this study showed that crucian carp is also a safe fish food, similar to other fish, because the indicators (THQ and HI) were below one. According to our knowledge, there are no acceptable standards in Poland regarding the contents of heavy metals studied in fish determined by us. Hence, it was decided to compare the marked levels of the elements with the Regulations of FAO (1983) and FAO/WHO (1989). On this basis, it was concluded that the levels of Cu, Zn, Fe and Mn in fish species were lower than the maximum levels. Nevertheless, research on the content of all heavy metals should be continued.

## Figures and Tables

**Table 1 ijerph-19-02307-t001:** Differences (x ± SD) in the contents of heavy metals (mg/kg wet weight) in the same organs of fish.

Species		Cu	Fe	Mn	Zn
Crucian carp (*Carassius carassius* L.)	x	0.77 a	6.95 a	0.12 b	19.52 a
	SD	0.06	1.19	0.02	5.14
Flounder (*Platichthys flesus* L.)	x	0.26 b	1.02 c	0.03 c	4.09 b
	SD	0.05	0.25	0.01	0.35
Gilthead seabream (*Sparus aurata* L.)	x	0.24 b	0.78 d	0.03 c	4.01 b
	SD	0.07	0.25	0.01	0.38
Mackerel (*Scomber scombrus* L.)	x	0.35 b	1.83 b	0.02 c	3.03 b
	SD	0.10	0.50	0.01	0.28
Blue grenadier (*Macruronus novaezelandiae* Hector)	x	0.36 b	1.16 c	0.03 c	2.98 b
	SD	0.08	0.19	0.00	0.15
Perch (*Perca fluviatilis* L.)	x	0.23 b	1.55 b	0.08 c	3.07 b
	SD	0.02	0.01	0.01	0.18
Rainbow trout (*Oncorhynchus mykiss* Walb.)	x	0.27 b	1.77 b	0.07 c	4.93 b
	SD	0.04	0.27	0.01	0.69
Tench (*Tinca tinca* L.)	x	0.23 b	1.65 b	0.09 c	3.33 b
	SD	0.05	0.41	0.02	0.43
Tilapia (*Oreochromis niloticus* L.)	x	0.16 c	1.06 c	0.05 c	3.05 b
	SD	0.03	0.24	0.02	0.32
Walleye pollock (*Gadus chalcogrammus* Pallas)	x	0.29 c	1.11 c	0.27 a	3.27 b
	SD	0.04	0.22	0.07	0.30

x—mean; SD—standard deviation; a, b, c and d—significant differences between the same organs of the different species (*p* < 0.05). The same letter indicates the absence of significant differences (*p* > 0.05).

**Table 2 ijerph-19-02307-t002:** Coverage of the demand (%) for Cu, Fe, Mn and Zn after consuming an average portion of fish (150 g).

Species	Cu						Fe						Mn						Zn					
	A	B	C	D	E	F	A	B	C	D	E	F	A	B	C	D	E	F	A	B	C	D	E	F
Crucian carp (*Carassius carassius* L.)	12.9	12.9	12.9	12.9	12.9	12.9	8.7	7.0	10.4	5.8	10.4	10.4	0.8	1.1	0.8	1.0	0.8	1.0	26.6	32.5	26.6	36.6	26.6	36.6
Flounder (*Platichthys flesus* L.)	4.4	4.4	4.4	4.4	4.4	4.4	1.3	1.0	1.5	0.9	1.5	1.5	0.2	0.3	0.2	0.3	0.2	0.3	5.6	6.8	5.6	7.7	5.6	7.7
Gilthead bream (*Sparus aurata* L.)	4.0	4.0	4.0	4.0	4.0	4.0	1.0	0.8	1.2	0.6	1.2	1.2	0.2	0.2	0.2	0.2	0.2	0.2	5.5	6.7	5.5	7.5	5.5	7.5
Mackerel (*Scomber scombrus* L.)	5.9	5.9	5.9	5.9	5.9	5.9	2.3	1.8	2.7	1.5	2.7	2.7	0.1	0.2	0.1	0.2	0.1	0.2	4.1	5.0	4.1	5.7	4.1	5.7
Blue grenadier (*Macruronus novaezelandiae* Hector)	5.9	5.9	5.9	5.9	5.9	5.9	1.4	1.2	1.7	1.0	1.7	1.7	0.2	0.3	0.2	0.3	0.2	0.3	4.1	5.0	4.1	5.6	4.1	5.6
Perch (*Perca fluviatilis* L.)	3.9	3.9	3.9	3.9	3.9	3.9	1.9	1.5	2.3	1.3	2.3	2.3	0.5	0.7	0.5	0.6	0.5	0.6	4.2	5.1	4.2	5.8	4.2	5.8
Rainbow trout (*Oncorhynchus mykiss* Walb.)	4.5	4.5	4.5	4.5	4.5	4.5	2.2	1.8	2.7	1.5	2.7	2.7	0.5	0.7	0.5	0.6	0.5	0.6	6.7	8.2	6.7	9.2	6.7	9.2
Tench (*Tinca tinca* L.)	3.8	3.8	3.8	3.8	3.8	3.8	2.1	1.6	2.5	1.4	2.5	2.5	0.6	0.8	0.6	0.7	0.6	0.7	4.5	5.6	4.5	6.2	4.5	6.2
Tilapia (*Oreochromis niloticus* L.)	2.7	2.7	2.7	2.7	2.7	2.7	1.3	1.1	1.6	0.9	1.6	1.6	0.4	0.5	0.3	0.4	0.3	0.4	4.2	5.1	4.2	5.7	4.2	5.7
Walleye pollock (*Gadus chalcogrammus* Pallas)	4.8	4.8	4.8	4.8	4.8	4.8	1.4	1.1	1.7	0.9	1.7	1.7	1.8	2.5	1.7	2.2	1.7	2.2	4.5	5.4	4.5	6.1	4.5	6.1

Explanation: A—boys (13–18 years old); B—girls (13–18 years old); C—man (19–50 years old); D—woman (19–50 years old); E—man (51–75 years old); F—woman (51–75 years old).

**Table 3 ijerph-19-02307-t003:** The hazard quotient calculated for metal contents in the muscle tissues of fish.

Species		Cu	Fe	Mn	Zn		
	THQ					TTHQ	HI
Crucian carp (*Carassius carassius* L.)		0.0112	0.0058	0.0005	0.0377	0.0551	
Flounder (*Platichthys flesus* L.)		0.0038	0.0008	0.0001	0.0079	0.0127	
Gilthead seabream (*Sparus aurata* L.)		0.0035	0.0006	0.0001	0.0077	0.0120	
Mackerel (*Scomber scombrus* L.)		0.0051	0.0015	0.0001	0.0059	0.0126	
Blue grenadier (*Macruronus novaezelandiae* Hector)		0.0052	0.0010	0.0001	0.0058	0.0120	0.164
Perch (*Perca fluviatilis* L.)		0.0033	0.0013	0.0003	0.0059	0.0109	
Rainbow trout (*Oncorhynchus mykiss* Walb.)		0.0039	0.0015	0.0003	0.0095	0.0152	
Tench (*Tinca tinca* L.)		0.0033	0.0014	0.0004	0.0064	0.0115	
Tilapia (*Oreochromis niloticus* L.)		0.0024	0.0009	0.0002	0.0059	0.0094	
Walleye pollock (*Gadus chalcogrammus* Pallas)		0.0040	0.0009	0.0011	0.0063	0.0125	
TDHQ		0.0456	0.0156	0.0032	0.0991		
RfD (mg/kg/day)		4.00 × 10^−2^	7.00 × 10^−1^	1.4 × 10^−1^	3.00 × 10^−1^		

RfD—Oral reference dose (mg/kg/day) [20]; THQ—Target Hazard Quotient; TTHQ—individual foodstuff; TDHQ—individual toxicant; HI—Hazardous Index.

**Table 4 ijerph-19-02307-t004:** Permitted levels of heavy metals in fish (mg/kg wet weight) described in the different literature.

References		Cu	Fe	Zn	Mn
FAO (1983)	[47]	30	-	30–150	-
FAO/WHO (1989)	[48]	30	-	40	2.5
Turkish Food Codex (TFC, 2002)	[49]	20	50	50	20
Anonymous (2005)	[50]	30	30	100	
Ministry of Agriculture, Fisheries and Food (MAFF, 2000)	[51]	20	-	50	-

## Data Availability

Not applicable.

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
