# Peer review of "Assessment of Fish Quality Based on the Content of Heavy Metals"

_ijerph, 2022, doi:10.3390/ijerph19042307_

Round 1

Reviewer 1 Report

Authors presented study was to estimate meat of fish: Crucian carp (Carassius Carassius L.), Flounder (Platichthys flesus L.), Gilthead seabream (Sparus aurata L.), Mackerel (Scomber scombrus L.), Blue grenadier (Macruronus novaezelandiae Hector), Rainbow trout (Oncorhynchus mykiss Walb.), Tench (Tinca tinca L.), Tilapia (Oreochromis niloticus L.), Walleye pollock (Gadus chalcogrammus Pallas) and Perch (Perca fluviatilis L.) on the content of toxic metals Cu, Fe, Mn and Zn. The results contained in the publication looked in the measurement of food monitoring and the state of the natural environment assessed on the basis of the analysis of fish meat in the food chain.

In this aspect, the publication is interesting and brings a lot of information about the quality of food that goes to our supermarkets. In terms of content, the paper is done correctly and the statistical calculations fully confirm the assumptions made.

An important aspect of the paper is the risk assessment of fish consumption in terms of the applicable legal regulations.

I recommend the publication in its present form for publication as it is important for a wide audience.

Author Response

Responses to the review

 Thank You for the review.

English language and style were checked.

Reviewer 2 Report

Manuscript ID: ijerph-1541358-peer-review-v1

Assessment of fish quality based on the content of heavy metals

I think it's a very interesting and very important topic in seafood context as regard quality assessment and presence of heavy metals of health concern.

The aim of this study was to estimate fish quality in terms of Cu, Fe, Mn and Zn contents in different fish species of commercial importance.

The topic is of interest for the academics and for human diet because of the importance of include fish in a balanced diet. There are few studies like this in literature: this study was original and innovative, well conduced, the research is well performed, the sampling analysis and statistical analysis were well done.

The conclusions are of interest and literature is update.

The manuscript is well written and easy to understand by readers. I believe that this manuscript does not need big changes but I think you can publish the manuscript after minor revision.

Author Response

(The authors gave the same response as above.)

Reviewer 3 Report

Generally this is well prepared manuscript worth to be published after some revision. At first author need to provide figure caption for a plate with studied fish illustrations with an appropriate explanation in text which part of fishes were used for analysis. Second, in 2.2 the authors need to clarify how the samples were prepared for element analysis (pls compare lines 85-89 with 89-93).

Further proposal for corrections:

lin 87 - replace "acid nitrogen" with "nitric acid"

Author Response

Responses to the review

All suggested corrections have been made.

Reviewer 4 Report

The manuscript with ID (ijerph-1541358) studied the analysis of heavy metal contents in the muscular tissues of several fish species. The manuscript is interesting and present in the scope of the journal; however, there are several revisions to be carefully revised by authors before the manuscript considered for publication in International Journal of Environmental Research and Public Health.

One of the problems of this manuscript is absence of the permissible limits of the measured heavy metals in fish tissues. My advice to the authors to add a table for the permissible limits of all studied metals in fish tissues and every number should be supported with an appropriate reference. This information will be helpful to support your findings. I think this manuscript will not be acceptable for readers without this info.

Minor revisions: -

Line 16: write full abbreviation

Line 19: How the authors confirm that all the studied fish species do not pose a threat to humans from metals? – Add response in the abstract

Line 20: “whose values below one” – you should add more details.

Line 22: “factors that may cause their growth” – This sentence was uncompleted.

Keywords: I think abbreviations in keywords are not required. Please revise

Line 66: All scientific names should be written italic – Please follow the comments in the attached PDF file.

Line 160 and after: Do not repeat the fill scientific names after their first appearance in the manuscript.

Line 283: remove “which feeds on invertebrate fauna,”

Line 414: Ctenopharyngodon idella and Perca fluviatilis should be written italic.

Author Response

Responses to the review

All suggested corrections have been made.

1.One of the problems of this manuscript is absence of the permissible limits of the measured heavy metals in fish tissues. My advice to the authors to add a table for the permissible limits of all studied metals in fish tissues and every number should be supported with an appropriate reference. This information will be helpful to support your findings. I think this manuscript will not be acceptable for readers without this info.

Reply: Table 4 was added

2. Line 16: write full abbreviation

Reply: Has been corrected

3. Line 19: How the authors confirm that all the studied fish species do not pose a threat to humans from metals? – Add response in the abstract

Reply: Has been corrected

4. Line 20: “whose values below one” – you should add more details.

Reply: Has been corrected

5. Line 22: “factors that may cause their growth” – This sentence was uncompleted.

Reply: Has been corrected

6. Keywords: I think abbreviations in keywords are not required. Please revise

Reply: Has been corrected

7. Line 66: All scientific names should be written italic – Please follow the comments in the attached PDF file.

Reply: Has been corrected

8. Line 160 and after: Do not repeat the fill scientific names after their first appearance in the manuscript.

Reply: Has been corrected

9. Line 283: remove “which feeds on invertebrate fauna,”

Reply: Has been corrected

10. Line 414: Ctenopharyngodon idella and Perca fluviatilis should be written italic.

Reply: Has been corrected

Reviewer 5 Report

Generally, the importance of the presently reported study is not questionable. However, it contains serious methodological problems which should be reconsidered. 
The number of the specimens: If authors bought fish from the local market, which is the easiest way of getting the material, then why they used not the same number of specimens. Making comparisons among different species based on data that do not result from the same number of specimens are not appropriate.
The origin of the specimens: It is also very important. Authors should define the origin of fishes (water bodies). Because due to the market, it is also possible that each specimen is from a different water body. In this case, the whole results are not appropriate for making these conclusions. 
Metals: investigation of Cu, Fe, Mn, and Zn is important, but it is not enough for making these conclusions. Especially, when the authors described, that these fish are not hazardous for human health. For this conclusion, they have to analyze at least the most important 10-12 metals (check the literature data), including at least the toxic heavy metals regulated by European Union (Cd, Pb; Hg is not a must-have, because Hg needs special methodology). 
The whole discussion part is just an extensive collection of literature data. I think authors should highlight their results according e.g. origin, and make some comparisons based on ecological roles of species. Especially, that they investigated both fresh and saltwater species. 

Specific comments: 

Title page: Please fill up the left side of title page.

Abstract: correct the scientific names (e.g. Carassius carassius), at O. mykiss, do not abbreviate the author. 
Line 19: delete the first ( sign
Define that the concentrations are for wet or dry weight.

Introduction:
Line 30-31: correct the https… link as citation
Line 48-50: the specific diet of age groups of a given species, and the bioindicator capacities of different age groups are also important. I recommend reading and cite the following article: https://www.sciencedirect.com/science/article/abs/pii/S1470160X19300019

DOI: 10.1007/s10661-017-5944-0 DOI: 10.1007/s11356-017-8606-4

Material & methods: 
Line 67-72, revise the scientific names, and at first, mention, write the author and date too for each species. Species names have to be in italics. 
Line 76: Delete this picture, and also it has not got any title or description.

Line 84-93: Cite relevant articles to the methodology. 

Results: Please add the results of the Shapiro-Wilk normality test. Did the authors perform the Levene test for checking the homogeneity of variability? Because if they did not, and the data has no homogeneous variability, you can not use ANOVA, you have to use non-parametric tests like Kruskal-Wallis analysis. 

Author Response

(The authors gave the same response as above.)

Round 2

Reviewer 4 Report

The authors have properly addressed the comments required. The manuscript now merits acceptance

Author Response

 Thank You for the review.

Best regards,

Joanna Łuczyńska

Reviewer 5 Report

Abstract: If you mention the author for species names, add the description date too at the first mention. Furthermore, the English name of species have not to start with capital letters all the time. Check fishbase.org and revise all of English names. Correct Carassius carassius. The second name cannot be started with capitals.

By the way, are the authors sure that they investigated Carassius carassius? Because it is very rare now in Europe, and it often has hybrids with Carassius gibelio. Can the authors show pictures of fish that were identified as Carassius carassius? I am afraid, that they were C. gibelio.

Abstract: 17-20: Please make these sentences as a comparison. It was significantly higher than…?

Line 20: This sentence about wet weight basis should be before the results.

Authors should refine the abstract and the discussion too. Because if they just investigated 4 metals, then they cannot say that it is consumable without any harmful effects. It just should be said, that these 4 metals did not exceed the maximum allowable concentrations etc.

Lines 47-50: Other organic pollutants like pesticides can be harmful to humans by the consumption of fish. I think authors should mention them too because pesticides are one of the most serious threatening organic pollutants of aquatic ecosystems: Authors should mention these aspects too
e.g.
https://www.mdpi.com/2071-1050/12/23/10152
https://www.mdpi.com/2305-6304/9/6/125

Lines 57-63: Authors should firstly mention that these metals are belonging to microelements, what microelements are, what kind of positive and essential effects they have. Not only the negative effects (when their concentrations are too high).

Lines 66-70: I recommend comparing the concentrations against the maximum allowable concentrations set by FAO (1983).

Lines 73: Correct the scientific names according to the fishbase. (e.g., its Carassius carassius!)

Line 94: This picture had not got a title, but I think the authors should delete it.

Line 97: Define the method of dissecting fish. What kind of types of equipment were used?

Line 151: It is Levene’s test.

Lines 150-154: Add which software was used for Shapiro-Wilk, Levene, and Kruskal-Wallis test. What was the post hoc test of Kruskall-Wallis?

Authors should also discuss the possible causes of these differences among the species. E.g. ecological needs, diet etc.

Also, it is an important question from where were fish caught? The markets should give this information and authors also should describe the origin of fish. Because it is the most important, that fish from which waterbodies can be eaten safely? Or they were harvested from aquaculture? Please describe it precisely species by species. It is a very important question. 

Author Response

Answers to the Reviewer's questions

Abstract: If you mention the author for species names, add the description date too at the first mention. Furthermore, the English name of species have not to start with capital letters all the time. Check fishbase.org and revise all of English names. Correct Carassius carassius. The second name cannot be started with capitals.

As suggested by the Reviewer, all comments in the Summary have been taken into account.

By the way, are the authors sure that they investigated Carassius carassius? Because it is very rare now in Europe, and it often has hybrids with Carassius gibelio. Can the authors show pictures of fish that were identified as Carassius carassius? I am afraid, that they were C. gibelio.

It was definitely a crucian carp (Carassius carassius Linnaeus, 1758), not a Prussian carp (Carassius gibelio Bloch, 1782), which are easily distinguished by body color and shape.

A photograph of the common crucian carp which was included in the research in this work.

Abstract: 17-20: Please make these sentences as a comparison. It was significantly higher than…?

As suggested by the Reviewer, this sentence has been changed

Line 20: This sentence about wet weight basis should be before the results.

As suggested by the Reviewer, the sentence “The results are expressed on a wet weight basis” is inserted in the text before the results, while in the Abstract it was mentioned against the first result given in parentheses.

Authors should refine the abstract and the discussion too. Because if they just investigated 4 metals, then they cannot say that it is consumable without any harmful effects. It just should be said, that these 4 metals did not exceed the maximum allowable concentrations etc.

 As suggested by the Reviewer, we hope that the changes concerning this issue have been applied correctly.

Lines 47-50: Other organic pollutants like pesticides can be harmful to humans by the consumption of fish. I think authors should mention them too because pesticides are one of the most serious threatening organic pollutants of aquatic ecosystems: Authors should mention these aspects too
e.g.
https://www.mdpi.com/2071-1050/12/23/10152
https://www.mdpi.com/2305-6304/9/6/125

 have the ability to accumulate in their tissues trace elements, heavy metals, pesticide residues and persistent organic pollutants in their tissues, including polychlorinated biphenyls (PCBs) [11-13]. These harmful substances reduce the health properties of fish meat.

As suggested by the Reviewer, literature was introduced (Look: Lines 51-60).

Lines 57-63: Authors should firstly mention that these metals are belonging to microelements, what microelements are, what kind of positive and essential effects they have. Not only the negative effects (when their concentrations are too high).

As suggested by the Reviewer, the order has been changed and the information listed has been added (Look: Lines 68-79).

Lines 66-70: I recommend comparing the concentrations against the maximum allowable concentrations set by FAO (1983).

As suggested by the Reviewer, the permitted levels of heavy metals in fish (mg/kg wet weight) described in different literature were inserted into the discussion in the form of a Table 4, and the test results were compared with these values ​​in the text (Look: lines 297-304).

Lines 73: Correct the scientific names according to the fishbase. (e.g., its Carassius carassius!)

As suggested by the Reviewer, the names have been corrected.

Line 94: This picture had not got a title, but I think the authors should delete it.

As suggested by the Reviewer, after accepting all changes, you will see that the drawing has been deleted.

Line 97: Define the method of dissecting fish. What kind of types of equipment were used?

As suggested by the Reviewer, the data has been entered (Look: lines 115-116).

Line 151: It is Levene’s test.

As suggested by the Reviewer, the name was changed.

Lines 150-154: Add which software was used for Shapiro-Wilk, Levene, and Kruskal-Wallis test. What was the post hoc test of Kruskall-Wallis?

As suggested by the Reviewer, the data has been entered.

Authors should also discuss the possible causes of these differences among the species. E.g. ecological needs, diet etc.

As suggested by the Reviewer, in the Discussion Part, the literature data concerning the suggested fragment was introduced.

Also, it is an important question from where were fish caught? The markets should give this information and authors also should describe the origin of fish. Because it is the most important, that fish from which waterbodies can be eaten safely? Or they were harvested from aquaculture? Please describe it precisely species by species. It is a very important question. 

As suggested by the Reviewer, the data has been entered (Look: Lines 100-104).

Best regards,

Joanna Łuczyńska
